# Nature-Derived Polysaccharide-Based Composite Hydrogels for Promoting Wound Healing

**DOI:** 10.3390/ijms242316714

**Published:** 2023-11-24

**Authors:** Hyerin Lee, Yerim Jung, Nayeon Lee, Inhye Lee, Jin Hyun Lee

**Affiliations:** School of Bio-Convergence Science, College of Biomedical & Health Science, Konkuk University, Chungju 27478, Republic of Korea

**Keywords:** natural polysaccharide, composite hydrogel, wound healing, therapeutic agent

## Abstract

Numerous innovative advancements in dressing technology for wound healing have emerged. Among the various types of wound dressings available, hydrogel dressings, structured with a three-dimensional network and composed of predominantly hydrophilic components, are widely used for wound care due to their remarkable capacity to absorb abundant wound exudate, maintain a moisture environment, provide soothing and cooling effects, and mimic the extracellular matrix. Composite hydrogel dressings, one of the evolved dressings, address the limitations of traditional hydrogel dressings by incorporating additional components, including particles, fibers, fabrics, or foams, within the hydrogels, effectively promoting wound treatment and healing. The added elements enhance the features or add specific functionalities of the dressings, such as sensitivity to external factors, adhesiveness, mechanical strength, control over the release of therapeutic agents, antioxidant and antimicrobial properties, and tissue regeneration behavior. They can be categorized as natural or synthetic based on the origin of the main components of the hydrogel network. This review focuses on recent research on developing natural polysaccharide-based composite hydrogel wound dressings. It explores their preparation and composition, the reinforcement materials integrated into hydrogels, and therapeutic agents. Furthermore, it discusses their features and the specific types of wounds where applied.

## 1. Introduction

The skin is the largest organ of our body, serving significant functions such as protecting our body from the external environment, bacteria, or pathogens, regulating body temperature, sensing external stimuli, and producing vitamin D [1]. Its wounds commonly occur in everyday life since it is the outermost organ. Everyday wounds or acute wounds can become chronic, leading to infections and complications when left untreated properly [2]. Effective wound care is crucial in preserving and improving overall human health and aesthetics, which are among the primary concerns of modern people. It is essential in surgical procedures, severe trauma, acute injury, burns, infections, and chronic wounds in patients with diabetes, vascular disease, immunodeficiency, and malignant degeneration [3,4].

Skin wound healing typically involves four physiological stages in sequence (Figure 1): hemostasis, inflammation, proliferation, and remodeling stages [5,6]. At the hemostasis stage, the constriction of blood vessels initially occurs, reducing blood loss and stopping bleeding. Then, platelets in the blood adhere to the site and aggregate around the wound to initiate coagulation. At the inflammation stage, inflammatory cells, such as neutrophils and macrophages, move to the wound site. Activated macrophages secrete tumor necrosis factor (TBF)-α. Interleukin (IL)-1β promotes the proliferation of fibroblasts and the expression of matrix metalloproteinase (MMP). Releasing various signaling molecules, such as cytokines and growth factors, promotes tissue repair. Moreover, the inflammation stage helps remove dead cells and potential pathogens. At the proliferation stage, fibroblasts move to the wound area and synthesize collagen to strengthen the healing tissue. Also, new blood vessels are formed (angiogenesis), enhancing the supply of nutrients and oxygen to the healing site. Epithelial cells around the wound proliferate and migrate to cover the wound. Over time, the wound gradually contracts, and granulation tissue forms. Finally, the wound undergoes maturation and remodeling at the skin tissue remodeling stage, where the collagen fibers realign. Through the healing stages, some wounds can result in complete restoration of tissue structure and function, while others may lead to scar formation. The wound-healing process and time are affected by several factors. These include age, nutrition, underlying health conditions, infection state, dressing types, and dressing application methods.

Although acute wounds generally heal in about two weeks, ongoing research has focused on achieving effective healing by reducing patient pain and discomfort, shortening the healing time, minimizing granulation tissue formation, and restoring the wounds to their original condition [7,8]. The methods to treat injuries and wounds vary depending on the patient’s condition and the characteristics and types of the wounds, considering the need to keep wounds clean and protected, support the immune system, and supply proper nutrition and hydration [9,10]. Given that most injuries result in bleeding, the initial step is to achieve adequate hemostasis, followed by appropriate cleaning and disinfecting of wounds to prevent infection. Subsequently, suturing or dressing is performed on wounds. Wound dressings, which originated as a clay plaster in ancient times, are used to cover wounds, protect them from foreign substances, and absorb exudates [10]. Over time, they have evolved to enhance wound treatment and healing. Generally, wound dressings can be categorized into dry and wet ones, depending on their moisture content. Although traditional dry wound dressings, like gauze, are used to cover wounds and protect against infection, they tend to adhere to the wound bed, causing pain upon their removal. In contrast, wet dressings, creating a moist wound environment, help not only relieve pain but also prevent drying and foster optimal conditions for wound healing and skin tissue regeneration [11]. One notable example of wet dressings is hydrogel dressings.

Hydrogels are materials possessing a three-dimensional network structure and a network chain composed of predominantly hydrophilic molecules. They have attained extensive attention in biomedical and pharmaceutical fields due to their biocompatibility, ability to retain high water content, tunability of structure and properties, ability to facilitate the loading of inclusions, and similarity to the physical properties of extracellular matrix [12,13]. They are classified as chemical, physical, and topological hydrogels based on the type of molecular bonding and structure forming the network crosslinks. They do not dissolve in water. Instead, they swell up to an equilibrium state. This swelling state is known to be governed by a thermodynamic balance between the mixing energy of chains in the network and solvent and the elastic strain of chains [14]. Their swelling behaviors depend on their crosslinking density (determining pore size), hydrophobicity, composition, and sensitivity to various external factors, such as temperatures, pHs, lights, sounds, and electric or magnetic fields [15,16]. In addition to these features, their remarkable capacities to absorb abundant wound exudate, maintain a moist environment, provide cooling and soothing conditions, and facilitate oxygen and nutrient migration enable hydrogels to be widely used for wound care and tissue regeneration [17,18]. Nevertheless, the mechanical properties of conventional hydrogels are weak because of high water content and local stress concentrated by structural heterogeneity, which limits their practical application [16].

This limitation can be effectively addressed by carefully selecting proper components and employing a composite hydrogel prepared by incorporating additional elements, such as nano- or micro-sized particles, fibers, fabrics, or foams within the hydrogel substrate [15,19]. Incorporating additional components enhances the functional properties of the hydrogel system, although composite hydrogels generally reduce the degree of swelling. They exhibit enhanced mechanical strength and structural integrity, controllable swelling, degradation, agent release and delivery, and sensitivity to external factors mentioned above [16,20]. When used as wound dressing, additional components can reduce the risks of infection and inflammation in wound dressing and reinforce hydrogel scaffolds to support cell growth and tissue regeneration. Consequently, composite hydrogel dressings (CHDs) promote wound healing and treatment by offering various added functionalities, including sensitivity to external factors, adhesiveness, mechanical strength, antioxidant and antimicrobial properties, tissue regeneration behavior, and control over the release of active therapeutic agents [21,22].

Hydrogels can be categorized as natural or synthetic based on the origin of the main components of the hydrogel network. Although the selection between natural or synthetic polymer-based hydrogels depends on the specific applications and favorable properties, natural polymer hydrogels may be preferred over synthetic hydrogels due to their biocompatibility, safety, sustainability, and degradability [17,20,21]. For these reasons, various research on advanced natural polymer-based hydrogel systems has been carried out for various medical applications, including wound dressings. Among natural polymers, polysaccharides are the most abundant in nature, eco-friendly, and sustainable, providing excellent biocompatibility, non-toxicity, and low-cost pricing. For these reasons, they have been extensively used in various bioindustries and continue to be researched for further advancements [23,24].

In this article, we have reviewed recent five-year research on natural polysaccharide-based CHDs for wound healing. Our focus was on advancements in cutting-edge composite hydrogel systems for effective wound healing. Additionally, we explored the primary components of the hydrogels, the reinforcements (nano- or micro-sized particles, fibers, woven or non-woven fabrics, or foams) incorporated to enhance their features, and the therapeutic agents (ThAs) added to promote wound healing actively. Furthermore, we have made a table summarizing the compositions, distinctive features, and applied wound types for each polysaccharide composite hydrogel system discussed in this review. Finally, we have addressed current challenges that need to be overcome, in addition to providing a comprehensive summary of the review.

## 2. Natural Polysaccharide-Based Composite Hydrogel Dressings (CHDs)

Polysaccharides are the most abundant biomaterials in nature, playing crucial roles in building cell walls and storing energy. Natural polysaccharides are carbohydrates originating or derived from natural resources, such as plants, animals, and microorganisms (Figure 2). They are long-chain biopolymers formed with monosaccharide units connected with glycosidic linkages [25]. Polysaccharides containing all the same types of monosaccharides are known as homopolysaccharides, while those composed of different types of monosaccharides are heteropolysaccharides [26]. Although the repeating unit monosaccharide is the same form for all the homopolysaccharides, they have distinct chemical structures depending on the position of glycosidic linkage or the existence of a branch, providing unique functions and physical properties like water solubility. Polysaccharides are typically classified into storage and structural polysaccharides. Representative energy-stored polysaccharides include starch and glycogen, while examples of structural polysaccharides are cellulose and chitin.

Polysaccharides, being abundant, sustainable, and eco-friendly, hold great promise as biodegradable biomaterials. They are essential components of living organisms and exhibit various biological activities, including cell adhesion and molecular recognition [27]. They can be continuously obtained from renewable resources, lowering the depletion of finite resources [28]. Their versatility, natural abundance, biodegradability, biocompatibility, and unique biological functions have made them widely researched and commercially used, particularly in the medical, pharmaceutical, cosmetic, and food industries. Their applications are drug delivery systems, tissue engineering scaffolds, carriers for cells and growth factors in regenerative medicine, and wound dressings. [23,24,29,30,31].

Polysaccharide-based hydrogels with a network have obtained significant attention as biomaterials in a wide range of applications. When used as wound dressings, their high biocompatibility reduces the risk of toxicity, adverse reactions, or inflammation when used in contact with biological tissues [32]. Additionally, their biodegradability prevents accumulation in the body, reduces the need for removal procedures, and minimizes patient discomfort [33]. Also, polysaccharide-based composite hydrogels, fabricated by adding reinforcements into the hydrogels, typically enhance their features. The following overview presents the advanced development of natural polysaccharide-based composite hydrogels in three forms (Figure 2) and their characteristics and applications for promoting wound healing (Figure 3). This discussion focused on the most popular and promising nature-derived polysaccharides utilized as biomaterials in various bioindustries: starch (St), glycogen (Gly), cellulose (Cel), chitosan (CS), sodium alginate (SA), agarose (AG), and hyaluronic acid (HA). Their chemical structures are depicted in Figure 4.

### 2.1. Starch (St)-Based CHDs

Starch, the most common and abundant carbohydrate in a granule form in plants, including rice, potatoes, and corn, is known as an energy-storage polysaccharide. It consists of many monosaccharide repeating units linked by β−glycosidic linkages. Starch comprises two main components: amylose and amylopectin. Amylose is a linear starch, contributing to its rigidity and low water solubility. In contrast, amylopectin has a branched structure, exhibiting a relatively higher water solubility. Serving as a primary energy storage substance in a branched form, it is prevalent in grains (rice, wheat, or corn), potatoes, and root vegetables [34]. Heating starch in water induces gelatinization, a process where starch granules absorb water, swell, break down the granular structure, and subsequently form a gel-like structure [35]. The physical gel formation imparts humidity-sensitive shape memory behavior [36]. Compared to other biopolymers, starch is abundant and cost-effective; thus, it has been used in various biofields [37,38]. Whether chemically or physically fabricated, starch-based hydrogels are utilized in various bioindustrial fields, encompassing biomedicine, pharmacy, and food [35,39].

Yang et al. [30] developed an injectable starch-based adhesive hydrogel, CoSt hydrogel, by incorporating dopamine-modified collagen, aldehyde-modified starch, and CaCO_3_ molecules. The CoSt composite hydrogel network formed through ester bonds, hydrogen bonds, and electrostatic interactions among its components, demonstrating strong adhesiveness to wet tissues (62 ± 4.8 KPa), self-healing behavior, shape adaptability, and excellent sealing performance (153 ± 35.1 mmHg). Furthermore, the composite hydrogels displayed significant in vivo hemostatic properties in rat liver and abdominal aorta injury models and faster wound healing within just one week in the rat skin incision model. This CoSt hydrogel, inspired by the adhesion mechanism of mussel, showed a promising potential as an effective adhesive. Nezami et al. [40] fabricated pH-sensitive and magnetic starch-based nanocomposite hydrogels (Fe_3_O_4_@St-IANCH) were manufactured via the polymerization of itaconic acid-modified starch, with the addition of Fe_3_O_4_ magnetic nanoparticles (MNP) serving as reinforcement and Guaifenesin (GFN) as a model drug. The more significant release of GFN molecules was observed at higher pH levels, attributed to the more considerable swelling of the hydrogels. The Fe_3_O_4_@St-IANCHs maintained biocompatibility and exhibited increased GFN release under the influence of a magnetic field. Compared to the hydrogels without MNP, those with MNP showed excellent in vivo wound healing performance. In the studies conducted by Abdollahi et al. [41], a nanocomposite hydrogel based on starch, referred to as CMS@CuO hydrogel, was prepared by casting the mixture of sodium carboxymethyl starch (CMS) mixed with CuO nanospheres (diameter: 20–50 nm, ~4 wt%). Citric acid was employed as a crosslinking agent, forming ester linkages as network crosslinks. The sodium carboxymethyl starch (CMS) was previously synthesized using starch with monochloroacetic acid. The CMS@CuO hydrogel exhibited biocompatible, antioxidant, and antimicrobial properties, resulting in a slightly faster wound healing rate in animal tests than those without CuO.

### 2.2. Glycogen (Gly)-Based CHDs

Glycogen (Gly) is an alternative energy-storing polysaccharide found in the liver and muscles of animals and humans, playing a role in energy metabolism. When our body needs energy, the glycogen stored is broken down immediately and converted into glucose through glycogenolysis [42]. The rapidly released glucose into the bloodstream provides energy for essential physiological processes, such as muscle contractions and the regulation of blood sugar levels [43,44]. It consists of numerous monosaccharide repeating units connected through α−glycosidic linkages and exhibits a high branch structure resembling a dendritic formation [45]. This branching structure improves water solubility and accelerates metabolism, leading to quick breakdown [46] into individual glucose molecules [47]. Therefore, unlike fat, Gly is considered a short-term energy storage molecule. Additionally, Gly is not typically used for commercial industries, unlike starch and cellulose. Instead, it plays a critical role in maintaining homeostasis and supporting various physiological functions of the body [48]. Due to its abundance of hydroxyl groups, it readily forms its derivatives with modified chemical structures and internal patterns. Moreover, its dendritic-like structure enables it to load active ingredients like drugs and easily make network structures, forming hydrogels [49,50]. Furthermore, its mechanical properties, such as higher strength and self-healing behavior, can be enhanced with a composite hydrogel system prepared by adding reinforcements, such as metallic ions, nanoparticles, fabrics, or foams [46,51,52].

Hasanin et al. [46] fabricated a novel cotton bandage dressing for wound repair. The bandage was manufactured by impregnating a cotton pad with a nanocomposite hydrogel through an ecofriendly green process. The CG@ZnONPs nanocomposite hydrogels were prepared with chitosan (here, denoted as C), glycogen (here, denoted as G), and ZnO NPs. The CG@ZnONPs-doped cotton pad exhibited antibacterial properties and enhanced thermal stability and mechanical properties. Compared to traditional gauze dressings, the nanocomposite hydrogel dressings displayed reduced inflammation, increased propagation of fibroblast cells, better tissue generation, and denser collagen deposition, resulting in flawless and faster wound healing. Meanwhile, Lan et al. [52] prepared the complex (GT/siMMP-9) particles of tetra (TETA)-conjugated Gly (GT) and matrix metalloproteinase 9 specific siRNA (siMMP-9) via electrostatic interactions. Subsequently, a PM(GT/siRNA) composite hydrogel was fabricated by encapsulating GT/siMMP-9 complex particles in a PF hydrogel composed of Pluronic F-127 (PF) and methylcellulose (MC). This composite hydrogel exhibited good biocompatibility and demonstrated sol–gel transition at body temperature (~37 °C). This thermosensitive gelation allows for use in arbitrary wound shapes. Furthermore, its sustained and localized release of siMMP-9 (up to 7 days) reduced MMP-9 expression, effectively improving wound healing in diabetic rats. Wu et al. [53] developed a composite hydrogel by encapsulating siMMP-9 enzyme proteins with bacterial cellulose-hyperbranched cationic polysaccharide (BC-HCP), referred to as BC-HCP/siRNA (or BC-HCP/si-MMP-9). The four HCPs used as gene carriers in this work were the fourth-generation polyamide-amine (PAMAM D4)-conjugated amylopectin (Amyp-D4), 3-(dimethylamino)-1-propylamine-conjugated amylopectin (Amyp-DMAPA), fourth generation polyamide-amine (PAMAM D4)-conjugated glycogen (Gly-D4), and 3-(dimethylamino)-1-propylamine-conjugated glycogen (Gly-DMAPA). The BC-HCP/siRNA composite hydrogels showed antibacterial properties and biocompatibility. The controlled release of siMMP-9 inhibited the MMP effect and enhanced wound healing for diabetic rats.

### 2.3. Cellulose (Cel)-Based CHDs

Cellulose (Cel) is the most abundant nature-derived organic material on Earth, contributing a considerable portion of plant biomass. It is a structural polysaccharide originating from the wall cell of plants, with a linear chain composed of many glucose units joined by β−glycosidic linkages. The unit has three different OH groups, which can be involved in various modification reactions, producing a variety of cellulose derivatives with other functional groups [54]. The features of cellulose derivatives depend on their type of substituents, degree of substitution, chemical structure, etc. Cellulose is also degradable, eco-friendly, moisture-resistive, thermal and acoustic insulating, and mechanically robust [55,56,57]. Consequently, it is widely used in commercial and industrial applications, such as papers, packing, pharmaceuticals, constructions, bioplastics, and biomedicines. Cellulose-based biomaterials are extensively used in biomedical and pharmaceutical applications, such as drug delivery systems [58], tissue engineering scaffolds [59], and wound dressings [60,61], owing to their hydrophilicity, biocompatibility, renewability, enhanced mechanical and barrier properties, and cost-effectiveness [62]. Indeed, cellulose-based composite hydrogels for wound care are extensively researched.

Mao et al. [63] developed a multifunctional rBC/MXene composite hydrogel for wound healing by incorporating regenerative bacteria cellulose (rBC), MXene (Ti3C2Tx), and epichlorohydrin (ECH) as a crosslinking agent via dual crosslinking. One crosslink was attributed to the chemical bonds between rBC and ECH, while hydrogen bonds and van der Waals interactions between rBC and MXene contributed to the other. Here, rBC refers to a regenerative form of cellulose produced by certain types of bacteria. MXenes are two-dimensional and electrically conductive compounds composed of transition metal carbides, nitrides, or carbonitrides. The hydrogels demonstrated effective wound healing under external electrical stimulation (ES) because a physical electric signal enhanced the proliferation activity of a fibroblast cell line, NIH/3T3. These hydrogels exhibited biocompatibility and a pore size of 100–500 μm, making them suitable for application in tissue engineering. Moreover, their wound healing efficacy in a full-thickness skin defect model was comparable to that of a commercial film dressing, Tegaderm. Meanwhile, Yeo et al. [64] developed methylcellulose (MC)-based composite hydrogels with tannic acid (TA) and Fe^3+^ for use in beauty devices or wound infection care using NIR lasers. The synthesis proceeded through a facile one-step method, and the hydrogel network was formed via hydrophobic interactions between methoxy groups in MC, hydrogen bonds between MC and TA, and coordination bonds between TA and Fe^3+^. The radiation of the NIR laser could control their gelation rate. They exhibited antibacterial, antioxidant, and UV-blocking properties due to multifunctional TA molecules. The TA-Fe^3+^ complex showed an excellent photothermal effect, which allowed the hydrogels to be used for beauty devices. TA release regulated via the content of Fe^3+^ ions enhanced antibacterial properties and infected wound healing. Wang et al. [65] fabricated an injectable, self-healing, near-infrared (NIR) photosensitive antibacterial composite hydrogel for accelerating wound healing. The hydrogel was formed with benzaldehyde-grafted carboxymethyl cellulose (CMCBA) and hydroxypropyl trimethyl ammonium chloride chitosan (HACC) via electrostatic interaction and Schiff base reaction. Then, polydopamine (PDA) molecules adhered to the surface of the CuS NPs and combined with curcumin molecules via π–π interactions, producing CuS@C NPs. The internal electric field formation between CuS and curcumin promoted the ROS generation to kill bacteria. They showed biocompatibility, photodynamic antibacterial properties, and excellent wound healing efficacy within 10 days.

### 2.4. Chitosan (CS)-Based CHDs

Chitosan (CS) is one of the most promising biomaterials due to its biocompatibility and antimicrobial properties. It is another structural polysaccharide deacetylated from chitin, a natural carbohydrate found in the shells of crustaceans like crabs and lobsters. Recently, CS can be obtained through shrimp-processing waste. Its repeating units contain -OH and -NH_2_ groups and are connected by β−glycosidic linkages. Its properties are dependent on the degree of acetylation (DA). The -OH and -NH_2_ groups are versatile for modification into various forms, allowing fine-tunable properties [66,67,68,69]. Protonated CS possessing positive charges (polycation) interacts with polyanions (e.g., heparin), thus forming physical gels [70]. Derivatives of CS, such as carboxymethyl CS (CMC) or carboxyethyl CS (CEC), showed improved solubility and additional interactions (indicating increased mechanical strength) and functionalities [68,71,72,73]. CS has been extensively used in biomedical and pharmaceutical realms due to its biocompatibility, mucoadhesive, chelating, antimicrobial, hemostatic, and biodegradable properties [74]. CS has been utilized for wound healing due to its ability to regulate drug release via manipulating interactions, accelerate tissue repair, and reduce inflammation [67,75,76,77].

Pan et al. [78] innovatively developed a CS-based dual bionic adhesive composite hydrogel inspired by the adhesion behavior of mussel and barnacle cement proteins. The hydrogel was prepared with catechol-conjugated chitosan (C-CS), tannic acid (TA), silk fibroin (SF), SA, and Ag NPs. Initially, the catechol groups of L-3,4-dihydroxyphenylalanine (L-DOPA) were first attached to CS molecules, leading to the preparation of CT-CS/TA/SF (C-CTS) and C-CTS/SA-Ag. Subsequently, the C-CTS/SA-Ag/dECM composite hydrogel was fabricated by incorporating a decellularized extracellular matrix (dECM) and the C-CTS/SA-Ag, showing antibacterial properties due to the Ag NPs and higher swelling. The hydrogen bonds, cation–π interactions, and electrostatic interactions within the hydrogel network resulted in its extreme adhesion to wet tissues. The hydrogel-PVA sponge composites prepared exhibited high compressive strength (140.08 ± 5.15 MPa) and Young’s modulus (43.61 ± 7.24 kPa), good shape memory behavior, and effective blood-sucking performance. Tehrani et al. [79] reported an injectable, thermosensitive, and oxygen-generating hybrid hydrogel based on chitosan. H₂O₂-loaded polylactic acid (PLA) microparticles (diameter: 4.481 ± 1.8 μm) were fabricated using the double emulsion method and subsequently integrated into the chitosan-based hydrogels formed with β-glycerophosphate (β-GP). Additionally, the amniotic membrane (AM) was incorporated as a therapeutic agent to enhance wound healing. The H₂O₂-PLA/CS/β-GP composite hydrogels demonstrate both hemocompatible and antibacterial properties, indicating great potential for improving wound healing. He et al. [68] fabricated a conductive multifunctional nanocomposite hydrogel for photothermal therapy (PTT) in treating bacteria-infected skin wounds. The hydrogels were prepared with CEC, benzaldehyde-terminated PF127, and carbon nanotubes (CNTs) as a PTT agent. Then, moxifloxacin hydrochloride (antibiotics) was loaded in the hydrogels. This nanocomposite hydrogel, which exhibits self-healing, pH-sensitive, biocompatible, antibacterial, hemostatic properties, and outstanding mechanical stability, demonstrated significant efficacy in healing infected wounds in a full-thickness rat skin model. Zhao and Yuan [80] developed a multifunctional injectable hydrogel dressing named OCEN for treating diabetic wounds. The hydrogel was fabricated with oxidized chondroitin sulfate (OCS), CMC, and phenol red-modified ε-poly-L-lysine (EPL-PR), while incorporating chondroitin sulfate-modified selenium NPs (CS@SeNPs) and infinite coordination polymer nanomedicine (ICPs) in the OCEN hydrogel. The synthesized CS@SeNPs exhibited a spherical shape with a diameter of 100 nm. The OCEN hydrogels demonstrated self-healing, pH sensitive, antibacterial, and biocompatible properties, hemostasis, shape-adaptivity, excellent adhesiveness, free radical scavenging properties, and significant absorbance of wound exudate. Furthermore, the diabetic full-thickness wound-healing phase of the hydrogels could be visually monitored, showing their great potential as a candidate for treating chronic diabetic wounds.

### 2.5. Sodium Alginate (SA)-Based CHDs

Alginate is an abundant natural polysaccharide derived from seaweed and found in the cell wall of brown algae, consisting of β(1,4)-linked D-mannuronic acid (M) and α(1,4)-L-guluronic acid (G) as the repeating units [81]. Alginate exists in the form of salts, including calcium, magnesium, and sodium salts. Among them, sodium salt (sodium alginates, SA) dissolves in water and does not adhere to cells, making it widely used in biomedical and pharmaceutical fields. The salts of alginates undergo gelation in the presence of divalent cations like Ca^2+^ and Zn^2+^ or trivalent cations like Al^3+^ and Fe^3+^. Although the gelation is ion-dependent and the properties of resulting gels vary depending on the ions involved, the gelation mechanism, known as the “egg-box” model, is induced by Ca cations [82,83]. SA hydrogels or SA-modified hydrogels [84,85] as wound dressings are promising due to their excellent biocompatibility and high capacity to absorb wound exudate and maintain a moist environment.

Zhang et al. [86] developed a light-responsive SA-based composite hydrogel with BiOCl (BOC) and polypyrrole (PPy) ((BOC-PPy (BP), both are biocompatible conducting materials) nanosheets using Ca^2+^ released from CaCO_3_. The BP-SA nanocomposite hydrogels showed a relatively higher storage modulus (G’, ~15 kPa) than loss modulus (G”) and photoelectric and photothermal dual properties due to the conducting nanosheets. The electric signal, converted from white light applied to the nanocomposite hydrogels, induced the migration of human umbilical vein endothelial cells (HUVECs) and enhanced angiogenesis, promoting wound healing in mice. Additionally, their antibacterial rate of up to 99.1% was achieved under near-infrared light illumination. Thus, the PB-SA composite hydrogel under white light was suggested to be a good candidate dressing for clinical wound healing. Yang et al. [87] fabricated a microenvironment (UME)-responsive SA composite hydrogel crosslinked with silicon quantum dots (SiQDs), incorporating hydroxyapatites (nHA) nanoparticles (NPs), to achieve scarless memory repair of urethral injuries using 3D bioprinting. The addition of nHA NPs enhanced its mechanical strength, and the SiQDs response to laser produced moderate reactive oxygen species (ROS). The structural reconfiguration of the scaffold responded to the Ca^2+^ ions in urine, reducing the swelling and increasing the stiffness of the hydrogels. Moreover, their tunable configuration adjusted their degradation rate to match the rate of urethral regeneration without reducing cell necrosis. Furthermore, SiQDs promoted angiogenesis and the differentiation of added adipose tissue-derived stem cells (ADSCs) while reducing scar formation through the generation of ROS under laser irradiation. Li et al. [85] prepared an injectable SA-grafted dopamine (DP) (SD) hydrogel containing the polydopamine-Fe(III)-doxorubicin (PFD) NPs, prepared by loading doxorubicin (DOX) in the polymerized PF NPs, for melanoma treatment and skin wound repair. Here, DOX, an anticancer drug, was released depending on external pH and near-infrared (NIR) irradiation, and the PFD NPs converted light to heat to kill cancer cells. The PFD/SD composite hydrogels exhibited self-healing behavior, biocompatibility, high adhesiveness, antibacterial and antioxidant properties, excellent melanoma tumor suppression, and promoted epidermal formation and in vivo wound healing under NIR. Zhang et al. [84] fabricated SA-COS-ZnO composite hydrogels without a crosslinking agent for wound care. The aldehyde groups of SAs oxidized by NaIO_4_ interacted with the amino group of COS to form the SA-COS hydrogel. Then, ZnO NPs were synthesized and loaded within the hydrogel, and the loading showed minimal impact on the swelling of the hydrogel. The composite hydrogel provided controlled release of Zn^2+^, causing antibacterial activity, a low hemolysis rate of 1.3~2.4% (indicating high blood compatibility), a reasonable moisture vapor transmission rate (MVTR) of 682 g/m²/24 h, improved G’ (~2 kPa), high biocompatibility, and accelerated wound healing efficacy as shown in its hematoxylin and eosin (H&E) results.

### 2.6. Agarose (AG)-Based CHDs

Agarose (AG) is a polysaccharide found in the cell walls of red algae. It is a linear polymer consisting of the repeat units of D-galactose joined with 3,6-anhydro-L-galactopyranose in glycosidic linkages [88]. AG forms a gel through extensive hydrogen bondings when cooled down from its dissolution temperature. Notably, thermal hysteresis exists in the sol-to-gel transition. Its melting temperature differs from the gelling temperature [89,90]. The transition temperatures depend on the concentration of AG solution and the content of methylation [91]. AG has gained enormous attention and is extensively used for biomedical applications. AG-based composite hydrogel dressings are popular due to their characteristics, such as biocompatibility, a cooling effect, maintenance of a moist environment, and thermal hysteresis [92,93,94].

Deng et al. [92] presented an approach to photothermal treatment for bacterial wounds using an AG composite hydrogel containing tannic acid (TA)-Fe(III) NPs. The TA-Fe NPs were rapidly synthesized through TA-FE assemblies obtained via the Fe(III) chelation of TA within the AG network formed via hydrogen bondings simultaneously. The AG-TA-Fe (ATF) composite hydrogel exhibited a higher tensile strength (~58.5 kPa), good cell viability, in vitro antibacterial properties, Fe(III)-induced photothermal sterilization effects, and an excellent therapeutic effect for healing Staphylococcus aureus-infected wounds on the backs of mice under NIR laser irradiation. Huang et al. [93] developed a novel antibacterial and anti-inflammatory composite hydrogel by incorporating modified AG, carboxymethyl agarose (CMA), and Ag^+^ through ionic interactions. The CMA-Ag composite hydrogel exhibited excellent physiochemical properties, such as pH and temperature responsiveness, cytocompatibility, and hemocompatibility. It swells more in an acidic environment because of the loss of ionic interaction between deprotonated CMS molecules and Ag^+^ ions. Additionally, it accelerated wound healing with smooth epithermal tissue and skin regeneration, showing a therapeutic effect on wound infection. Eivazzadeh-Keihan et al. [94] developed a novel biocompatible nanocomposite hydrogel, named lignin–agarose/SF/ZnCr_2_O_4_, comprising of lignin-modified agarose, silk fibroin, and Zinc chromite (ZnCr_2_O_4_) NPs. The composite hydrogel displayed significant swelling (swelling %: 815 ± 14%) and high mechanical properties (elastic modulus: 29.51 ± 0.05 MPa and tensile strength: 176.2 ± 1.4 MPa). In addition, the G’ of composite hydrogel was higher than its G”, indicating that it possessed more elastic properties than viscous ones. Moreover, its G’ was higher than that of lignin–agarose/SF hydrogels without ZnCr_2_O_4_ NPs, indicating that ZnCr_2_O_4_ NPs contributed to the enhanced mechanical properties of the composite hydrogels. The composite hydrogel showed excellent hemocompatible, antioxidant, anti-infective, and antimicrobial properties and demonstrated positive in vivo assay results, achieving a fast wound healing time (5 days) in adult male mice.

### 2.7. Hyaluronic Acid (HA)-Based CHDs

Hyaluronic acid (HA) is a polysaccharide and non-sulfated glycosaminoglycan found in the extracellular matrix, connective tissue, body fluids, and lubricant fluids for joints, called mucopolysaccharides [95]. HA has a chemical structure in which d-glucuronic acid and d-N-acetylglucosamine molecules are alternatively linked with β−glycosidic bonds. Additionally, it contains -OH and -COOH functional groups [96]. It becomes negatively charged with -COO groups upon dissolving in water, making it highly hydrophilic and causing significant swelling. Its pharmacokinetics are well established, and its lifetime in the body circulation is several minutes. The biocompatible HA promotes the production of M2 phenotype macrophages, reducing inflammation and enhancing cell proliferation [97]. These features make it suitable for biomedical, pharmaceutical, and cosmetic applications [98,99]. HA- or functional group-modified HA-based hydrogel wound dressings with or without inclusions have also been extensively studied and developed [100,101].

Han et al. [102] developed a diagnostic and therapeutic HA-based composite hydrogel for diabetic wound healing. Vascular endothelial growth factor (VEGF) was encapsulated with poly(lactic-co-glycolic acid) (PLGA)-based nanobubles (PFH@VEGF-PLGA NBs) using double emulsion. These VEGF-loaded NBs were mixed with HA-NH_2_ to form the HA-NH_2_@PFH@VEGF-PLGA hydrogel. In addition, MnO_2_ peroxidase enzymes (GOx-MnO_2_) were synthesized with glucose oxidase (GOx) and MnO_2_. The enzymes were mixed with HA-CHO to fabricate the HA-CHO@MnO_2_@GOx hydrogel. Then, the injectable HA-NH_2_@PFH@VEGF-PLGA and HA-CHO@ MnO_2_@GOx hydrogels were combined to form the US@GOx@VEGF (UGV) composite hydrogel entrapping VEGF-loaded NBs and GOx-MnO_2_ enzymes via a Schiff base reaction, which was proceeded using ultrasonication. The controlled release of VEGF from the UGV composite hydrogel effectively enhanced its role in stimulating vascularization via sugar level reduction. The H_2_O_2_ produced during this process reacted with MnO_2_, generating Mn^2+^, an MRI indicator, consequently improving MRI performance. Furthermore, the UGV composite hydrogels exhibited remarkable self-healing and sound-responsive properties, monitored real-time blood sugar levels, and promoted diabetic skin wound healing in vivo using controlled VEGF release. Zhang et al. [103] reported the development of a photo-responsive HA hydrogel containing PLGA-nitrobenzene (NB) capsules with transforming growth factor-β (TGFβ), a signaling protein. While TGFβ is known to control tissue homeostasis and regeneration, promoting skin wound closure and scar treatment, its uncontrolled activation can cause side effects, including fibrosis, bone loss, and immunosuppression [104]. This work presents the light-modulated pulsatile release of TGFβ encapsulated in PLGA capsules to control its release, allowing the composite hydrogel to provide scarless wound healing in vivo tests. Lui et al. [97] fabricated a high molecular weight HA(HHA)-based composite hydrogel for controlled inflammation microenvironment to enhance chronic diabetic wound healing. The thioether-grafted high molecular weight HA (HHA-S) was electrospun to obtain nanofibers. The thioether showed the ability to scavenge the ROS, alleviating inflammation reactions. The HHA-S nanofibers were crosslinked by Fe^3+^ ions to obtain the FHHA-S/Fe composite hydrogel. A higher concentration of Fe^3+^ ions enhanced the mechanical stability of the composite hydrogel even in exudate environments and its G’. The HHA within the composite hydrogel network could enhance the transformation of assembled M1 macrophages into M2 phenotypes, accelerating the transition from the inflammatory stage to the wound remodeling state, thereby improving chronic wound healing of the composite hydrogel.

Finally, Table 1 presents a summary of the components, distinctive features, and applied wound types for the natural polysaccharide-based composite hydrogel systems discussed above.

## 3. Conclusions and Challenges

This article presents a comprehensive literature review on recent advancements in the development of natural polysaccharide-based composite hydrogel dressings for wound healing and tissue regeneration. The seven polysaccharides most attractive in biomedical and pharmaceutical applications, namely starch, glycogen, cellulose, chitosan, alginate, agarose, and hyaluronic acid, were regarded for this review. We specifically focused on their preparation methods, compositions, inclusions (particles, fibers, fabrics, or foam), characteristics, dressing-applied wound types, and therapeutic agents. We recognized the notable innovations in hydrogel wound dressings through the overview. In addition, we have observed that natural polysaccharide-based composite hydrogel dressings are a more effective solution for enhancing wound healing and treatment compared to traditional polysaccharide-based hydrogel dressings without reinforcements. The increased efficacy of these composite hydrogels in promoting wound healing can be attributed to the specific functionalities conferred by incorporating additional elements, complementing the features provided by traditional hydrogel dressings. The incorporated elements contribute to responsiveness to certain external factors, improved adhesiveness and mechanical robustness, controlled release of therapeutic agents, and excellent antioxidant and antimicrobial properties. Consequently, the most suitable natural polysaccharide-based composite hydrogel dressing needs to be designed and selected to promote the healing of a specific or chronic wound, considering various factors, such as wound type, exudate levels, and patient comfort.

Despite significant progress in natural polysaccharide-based composite hydrogel dressings, certain challenges remain for further research. Compared to synthetic polymer-based composite hydrogels, natural polymer-based ones exhibit relatively weak mechanical strength, the potential for allergenic reactions, less controlled morphology, degradation rate, swelling behavior, and higher costs due to expensive raw materials and extraction processes. Ensuring consistency, sterility, and an adequate shelf life poses ongoing challenges for these natural material-based wound dressings. Furthermore, continuous efforts are to reduce the influence of factors that can hinder therapeutic effectiveness, such as microbial infections and reactive oxygen species (ROS), which significantly impede the wound-healing process. We anticipate that ongoing and future research endeavors will yield innovative solutions to address these challenges.

## Figures and Tables

**Figure 1 ijms-24-16714-f001:**
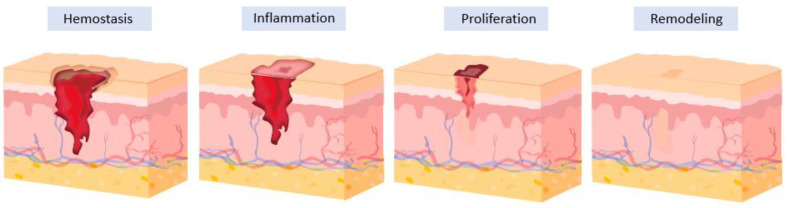
Schematic illustration of the four physiological stages in the skin wound-healing process: hemostasis, inflammation, proliferation, and remodeling of the skin tissue.

**Figure 2 ijms-24-16714-f002:**
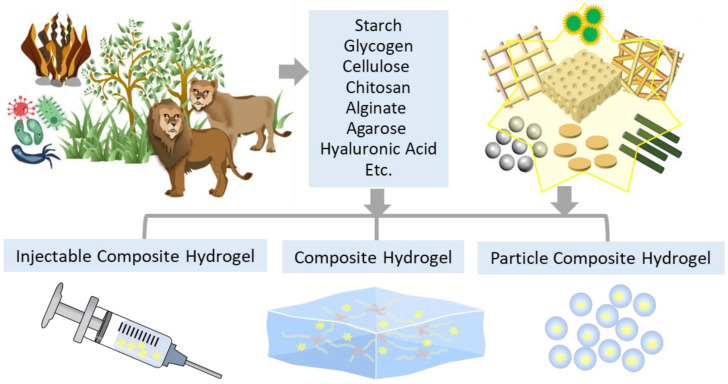
Schematic representation of three different forms of naturally derived polysaccharide-based composite hydrogels for wound dressing. Polysaccharides discussed in this work are listed. The yellow stars represent reinforcements (particles, fibers, non-woven and woven fabrics, or foams) and therapeutic agents added for fabricating composite hydrogels.

**Figure 3 ijms-24-16714-f003:**
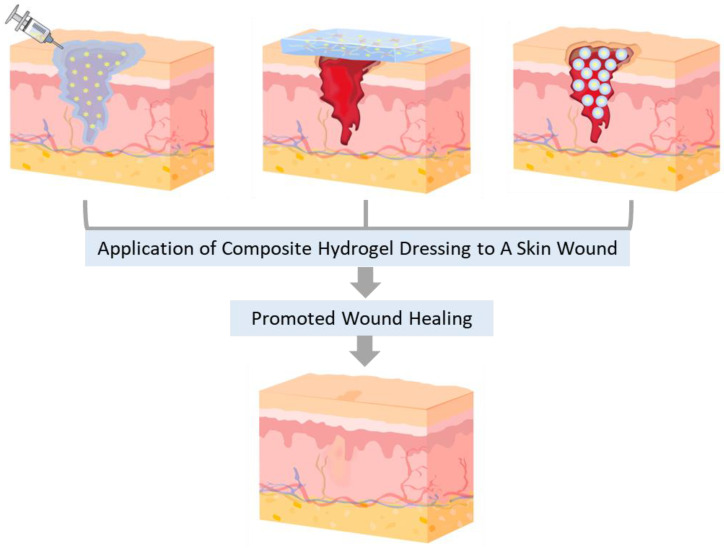
Schematic illustration of skin wound healing effectively promoted by applying composite hydrogels in different forms.

**Figure 4 ijms-24-16714-f004:**
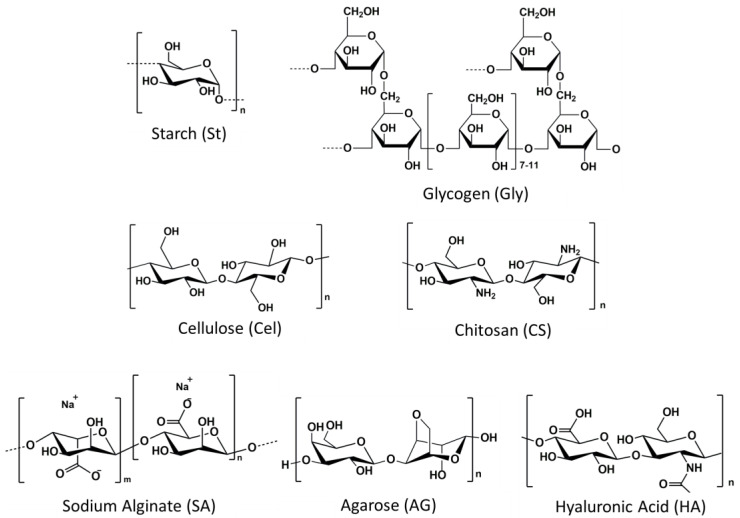
Chemical structures of starch (St), glycogen (Gly), cellulose (Cel), chitosan (CS), sodium alginate (SA), agarose (AG), and hyaluronic acid (HA).

**Table 1 ijms-24-16714-t001:** Composite hydrogel systems for wound healing and their components, features, and applications.

Name	Components	Features	Applications	Ref.
Starch (St)				
CoSt	Aldehyde-St, DP-conjugated Col(here, denoted as Co), CaCO_3_	Injectability, self-healing ability, shape adaptability, hemostatic efficiency, strong wet tissue adhesiveness (62 ± 4.8 Kpa), high sealing performance (153.2 ± 35.1 mmHg)., wound healing efficacy.	Emergency wounds, Non-pressing, hemostasis	[30]
Fe_3_O_4_@St-IANCH	IA-modified St, Fe_3_O_4_ MNPs (ThA: GFN)	pH-sensitive and magnetic response, cytocompatibility, controlled GFN release, and wound healing efficacy.	General wounds	[40]
CMS@CuO	Sodium CMS, CuO NPs	Solution casting for gel synthesis, biocompatible, antioxidant, and antimicrobial properties, and wound healing efficacy.	General wounds	[41]
Glycogen (Gly)				
CG@ZnONP	Gly(here, denoted as G), CS(here, denoted as C), ZnO NPs, Cotton pads	Nanocomposite, antibacterial properties, high thermal stability and mechanical properties, excellent epithelialization and tissue generation, lower inflammation, flawless wound healing.	General wounds	[46]
PM(GT/siRNA)	PF, MC, GT(Gly-TETA) (ThA: siMMP-9)	Modified Gly nanoparticles encapsulating siMMP-9, sol–gel transition behavior (G′ > G″ at body temperature), no-toxicity, shape adaptability, and diabetic wound healing improved through the sustained and localized release of siMMP-9.	Diabetic chronic wounds	[52]
BC-HCP/ siRNA (or BC-HCP/ siMMP-9)	BC, four HCP (Gly-DMAPA, Gly-D4, Amyp-DMAPA, Amyp-D4), (ThA: siMMP-9)	BC-HCPs as gene carriers, antibacterial properties, biocompatibility, and wound healing enhanced through the inhibition of MMP-9 by the controlled release of siMMP-9.	Diabetic wounds	[53]
Cellulose (Cel)				
rBC/MXene	rBC, MXene, ECH	Dual crosslinking (hydrogen bonding/van der Waals interaction and ECH crosslinking), EF-regulated wound healing, high surface roughness, wound healing efficacy.	General wounds	[63]
MC/TA/Fe	MC, TA, Fe^3+^ (ThA: TA)	Fast gelation, dual crosslinking (coordination/hydrogen bonds in TA/Fe and hydrophobic interactions in MC), pH and temperature sensitive, antibacterial, and antioxidant properties, photothermal and UV-blocking behavior, and wound healing efficacy.	Infected wounds, beauty devices	[64]
CMC/HACC	CMCBA, HACC, CuS@C (ThA: Curcumin)	Injectable, self-healing, EF-responsive, photocatalytic properties, excellent light-induced antibacterial activity, wound healing efficacy.	General wounds	[65]
Chitosan (CS)				
C-CTS/SA-Ag/dECM	CTS(C-CS/SF/TA), SA, Ag NPs, L-DOPA (ThA: dECM)	Robust wet-tissue adhesiveness (151.40 ± 1.50 kPa), fast multimodal self-healing ability, excellent antibacterial property, higher swelling, hemostatic efficiency, and wound healing efficacy.	Massive hemostasis, organ incision, deep wounds	[78]
H₂O₂-PLA/CS/β-GP	CS, H₂O₂-loaded PLA MPs, β-GP (ThA: AM, H_2_O_2_)	Injectability, oxygen-generating performance, hemocompatibility (hemolysis rate: <5%), thermosensitive and antibacterial properties, wound healing efficacy.	General wounds	[79]
CEC/PF/CNT	CEC, b-PF127, CNT (ThA: Mox)	Conductive, self-healing, hemostatic, and antibacterial properties, wound healing using photothermal therapy.	Infected wounds, hemostasis	[68]
OCEN	CMC, OCS, EPL-PR, CS@SeNPs, (ThA: ICPs)	Injectable, self-healing, and pH-sensitive properties, shape-adaptivity, excellent adhesiveness, antibacterial activities, biocompatibility, free radical scavenging properties, and large absorbance of wound exudate.	Diabetic wounds, hemostasis	[80]
Sodium Alginate (SA)				
BP-SA	SA, BP NSs	Light-responsive and antibacterial properties, Proper modulus (G’: ~15 kPa), wound healing efficacy.	General wounds	[86]
SA-nHA-SiQDs	SA-SiQDs, nHA NPs, Ca^2+^ (ThA: ADSCs)	UME-responsive 3D-printing, laser-activated ROS production, enhanced scaffold stiffness (G’: ~100 kPa), controlled degradation, scarless wound healing efficacy.	Urethral tissue repair	[87]
SD-PFD	SA-DP (SD), PFD NPs (ThA: DOX)	Injectable and self-healing behaviors, pH sensitiveness, temperature sensitiveness, excellent photothermal and antibacterial properties, adhesiveness, and wound healing efficacy.	Melanoma care	[85]
SA-COS-ZnO	Oxidized SA, COS, ZnO	Good MVTR, excellent blood compatibility, antibacterial and mechanical properties, and wound healing efficacy.	Scald wounds	[84]
Agarose (AG)				
ATF	AG, TA-Fe NPs	Good tensile strength (ATF-5: 58.5 kPa), superior photothermal sterilization effect, good biocompatibility, antibacterial activity, and wound healing efficacy.	Infected wounds	[92]
CMA-Ag	CMA (modified AG), Ag^+^ ions	Crosslinks by ionic interaction, pH and temperature responsiveness, antibacterial properties, biocompatibility, hemocompatibility, and wound healing efficacy.	Infected wounds	[93]
Lignin–AG/SF/ZnCr_2_O_4_	Lignin, AG, SF ZnCr_2_O_4_ NPs	Self-healing, high swelling (815 ± 14%), enhanced mechanical properties (elastic modulus: 29.51 ± 0.05 MPa and tensile strength: 176.2 ± 1.4 MPa), biocompatibility, antimicrobial, anti-infective, and antioxidant properties, hemocompatibility, fast wound healing time (5 days).	General wounds, tissue engineering	[94]
Hyaluronic acid (HA)				
US@GOx@VEGF (UGV)	HA, GOx, MnO_2_ PLGA, PFH, (ThA: VEGF, GOx-MnO_2_),	Injectable, self-healing, and sound-responsive properties, real-time monitoring of blood sugar levels, and wound healing promoted by controlled VEGF release.	Diabetic wounds	[102]
HA-NB/HA-CDH	HA, PLGA-NB (ThA: TGFβ)	Injectable and adhesive properties, nanobubbles (D: ~220 μm), scarless wound healing.	Diabetic wounds,	[103]
FHHA-S/Fe	HHA, Fe^3+^	Crosslinking of electrospun HA nanofibers with F^3+^ ions (D: ~60 nm), at higher Fe^3+^ ions, higher mechanical stability and G’, antibacterial property, wound healing efficacy.	Chronic diabetic wounds	[97]

Abbreviations: ADSCs: adipose tissue-derived stem cells, AG: agarose, AM: amniotic membrane, Amyp-D4: the fourth generation polyamide-amine (PAMAM D4)-conjugated amylopectin, Amyp-DMAPA: 3-(dimethylamino)-1-propylamine-conjugated amylopectin, ATF: AG/TA-Fe NPs, BC: bacteria cellulose, BC-HCP: bacterial cellulose-hyperbranched cationic polysaccharide, BP: BiOCl/Polypyrrolidone, b-PF127: benzaldehyde-terminated Pluronic F127, C-CTS: C-CS/TA/SF (C-CS: catechol-conjugated chitosan, TA: tannic acid, SF: silk fibroin), CDH: carbohydrazide, CEC: N-carboxyethyl chitosan, Cel: cellulose, CMA: carboxymethyl agarose, CMC: carboxylmethyl cellulose, CMCBA: benzaldehyde-grafted carboxymethyl cellulose, CMS: carboxymethylated starch, CNT: carbon nanotubes, Col: collagen, COS: chitosan oligosaccharide, CoSt: collagen/starch, CS: chitosan, CuS@C: CuS-grafted-curcumin, D: diameter, dECM: decellularized extracellular matrix, DOX: doxorubicin, DP: dopamine, ECH: epichlorohydrin, EF: electric field, FHHA-S: electrospun HHA-S nanofibers with Fe^3+^, G’: stroage modulus, GFN: Guaifenesin, Gly: glycogen, Gly-D4: the fourth generation polyamide-amine (PAMAM D4)-conjugated glycogen, Gly-DMAPA: 3-(dimethylamino)-1-propylamine-conjugated glycogen, GOx: glucose oxidase, GT: Gly- TETA(triethylenetetramine), HA: hyaluronic acid, HACC: hydroxypropyl trimethyl ammonium chloride chitosan, HHA: high moleular weight hyaluronic acid, IA: itaconic acid, IANCH: itaconic acid nanocomposite hydrogel, ICPs: infinite coordination polymer nanomedicine, L-DOPA: L-3,4-dihydroxyphenylalanine, MC: methylcellulose, MMP-9: matrix metalloproteinase 9, MNPs: magnetic nanoparticles, Mox: moxifloxacin hydrochloride, MVTR: moisture vapor transmission rate, Mxene: Ti_3_C_2_T_x_, NB: o-nitrobenzene, nHA: nanosized hydroxyapatites, NPs: nanoparticles, NSs: nanosheets, OCEN: OCS/CMC/EPL-PR/CS@SeNPs(OCS: oxidized chondroitin sulfate, CMC: carboxymethyl chitosan, EPL-PR: phenol red-modified ε-poly-L-lysine, CS@SeNPs: chondroitin sulfate-modified selenium nanoparticles), PF: Pluronic F127, PFD: polydopamine-Fe(III)-doxorubicin, PFH: perfluorooctane, PLA MPs: poly L-lactic microparticles, PLGA: poly-lactic-co-glycolic acid, PM: PF(Pluronic F127) and MC(methylcellulose) hydrogel, rBC: regenerated bacterial cellulose, ROS: reactive oxygen species, SA: sodium alginate, SD: SA-grafted DP (SA: sodium alginate, DP: dopamine), SF: silk fibroin, siMMP-9: MMP-9 specific siRNA, siRNA: small interference RNA (ribonucleic acid), SiQDs: Silicon quantum dots, St: starch, TA: tannic acid, TETA: triethylenetetramine, TGFβ: transforming growth factor-β, ThA: therpeutic agent, UGV: US@GO@VEGF, UME: urine microenvironment, US: ultrasound, UV: ultraviolet-visible, VEGF: vascular endothelial growth factor, ZnO: zinc oxide, β-GP: β-glycerophosphate, ZnCr_2_O_4_: zinc chromite.

## Data Availability

Not applicable.

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
