# Peer review of "Nature-Derived Polysaccharide-Based Composite Hydrogels for Promoting Wound Healing"

_ijms, 2023, doi:10.3390/ijms242316714_

Round 1

Reviewer 1 Report

Comments and Suggestions for Authors

The manuscript “Nature-Derived Polysaccharide-Based Composite Hydrogels for Promoting Wound Healing” from Lee et al. is an interesting review on current developments on wound healing with the use of natural polysaccharide composites. Overall, the review could be a good addition for the International Journal of Molecular Sciences, although some significant points should be addressed by the authors.

1)     Firstly, there should be a general reformatting of the manuscript regarding the English language. In some areas there the meaning is difficult to decipher. Such as:

-        Line 37 “tissue regeneration as a barrier”

-        Lines 36-56 please rewrite the wound healing procedure more clearly and with more details

-        Lines 80-84 please check the phrasing

-        Line 113 “variety of research” please rephrase

-        Lines 119-128 please make the paragraph more clear

-        Line 145 “The representative reserve” please rephrase

-        Lines 239-240 please improve the phrasing

-        Line 475 “can cause the risk of side effects, including fibrosis, bone loss” . Rephrase this

-        Line 497 “Specifically, we focused on preparing the composite hydrogel systems” Rewrite this sentence. You are describing the composite hydrogel systems in this review, you have not prepared and conducted the experiments.

2)     Line 76 “hydrophilic components”. Please define what are these components

3)     Line 93 please add the corresponding reference

4)     One or two additional pictures with schematics describing the mechanisms of wound healing for one or two mentioned polysaccharide composites would be a very good addition

5)     Figure 2 “The yellow stars: reinforcements (particles, fibers, non-woven and woven fabrics, and foams) and therapeutic agents.” what you are describing is unclear

6)     Please also elaborate more when describing the improvements of adding composite in the polysaccharide hydrogels. For example, for the starch based hydrogel (CMS@CuO hydrogel) you mention that it results in advanced wound healing performance. It would be better to give to the point quick explanations such as statistics of healing improvement or information on successful in-vivo experiments. Please also do that for the other polysaccharide examples.

7)     Lines 243-251 this research refers more to Osteogenic and cartilage differentiation which is more towards the tissue regeneration application while the other examples you are giving are focusing more on wound healing. Either remove this application or convey your message better.

8)     The numbers in chemical compounds should be in subscript or superscript format. Examples are MnO2 or HA-NH2 in lines 460 and 459

9)     Lines 467-468 “The H2O2 produced from GOx-MnO2 generated Mn2+ also contributed to improved MRI performance”. Elaborate more on how MRI performance was improved.

Comments on the Quality of English Language

I strongly suggest the authors get editing help from someone with full professional proficiency in English.

Author Response

The file is attached.

Reviewer 2 Report

Comments and Suggestions for Authors

The review manuscript from Lee and collaborators aimed to discuss the application of Composite Hydrogels fabricated using natural polysaccharides for wound healing. 
The authors have reviewed publications from last five years which report the development of composite hydrogels using 
sodium alginate (SA), agarose (AG), starch (St), glycogen (Gly), cellulose (Cel), chitosan (CS), and hyaluronic Acid (HA). 

Major issues

1- The authors should present the database(s) they used to search the articles.

2- In general the paragraphs are too long. Please consider to split them in two or more paragraphs, according to the subjects. 

3- Please justify your decision to discuss only formulation with the chosen polysaccharides.

4- Please provide the chemical structures for each polysaccharides discussed (including their derivatives).

5- The authors should provide more details about the findings of each manuscript enrolled in the manuscript. For instance is extremely important to make clear if each healing study was performed using animal models (and what model was used) or in humans.

minor issues are highlighted in the attached pdf file. 

Author Response

The file is attached.

Round 2

Reviewer 1 Report

Comments and Suggestions for Authors

The suggestions were correctly applied by the authors in the manuscript. The review can be accepted in its current form.

Reviewer 2 Report

Comments and Suggestions for Authors

The authors have answered my questions.